# Measuring precipitation with a geolysimeter

Craig D. Smith[1], Garth van der Kamp[2], Lauren Arnold[1], Randy Schmidt[2]

[1] Environment and Climate Change Canada, Climate Research Division, Saskatoon, S7N 3H5, Canada

[2] Environment and Climate Change Canada, Watershed Hydrology and Ecology Research Division,
   Saskatoon, S7N 3H5, Canada

Correspondence to: Craig D. Smith (craig.smith2@canada.ca)

## Abstract

Using the relationship between measured groundwater pressures in deep observation wells and with total surface loading, a geological weighing lysimeter (geolysimeter) has the capability of measuring precipitation event totals independent of conventional precipitation gauge observations. Correlations between ground water pressure change and event precipitation were observed at a co-located site near

Duck Lake, SK over a multi-year and multi-season period. Correlation coefficients ($r^2$)s varied from 0.99 for rainfall to 0.94 for snowfall. The geolysimeter was shown to underestimate rainfall by 7% while overestimating snowfall by 9% as compared to the unadjusted gauge precipitation. It is speculated that the underestimation of rainfall is due to unmeasured runoff and evapotranspiration within the responsesensing area of the geolysimeter during larger rainfall events while the overestimation of snow is

at least partially due to the systematic undercatch common to most precipitation gauges due to wind. Using recently developed transfer functions from the World Meteorological Organization's (WMO) Solid Precipitation Intercomparison Experiment (SPICE), bias adjustments were applied to the Alter shielded, Geonor T-200B precipitation gauge measurements of snowfall to mitigate wind induced errors. The bias between the gauge and geolysimeter measurements was reduced to 3%. This suggests that the

geolysimeter is capable of accurately measuring solid precipitation, and can be used as an independent and representative reference of true precipitation.

## 1  Introduction

It is well recognized that it is difficult to accurately measure solid precipitation with an accumulating

precipitation gauge on account of the systematic undercatch due to wind (e.g. Sevruk et al., 1991;

Goodison et al., 1998; Kochendorfer et al., 2017). For example, it has been shown that the single Alter shielded Geonor T-200B precipitation gauge could underestimate snowfall by as much as 60% at average wind speeds (~5 m $s^{-1}$ at gauge height) on the Canadian Prairies (Smith, 2009). The World Meteorological Organization's (WMO) Solid Precipitation Intercomparison Experiment (SPICE) as described by Nitu et al.

(2012) and Rasmussen et al. (2012), has recently documented similar results with Alter shielded and unshielded gauges undercatching snowfall by an average of ~40% and ~60% respectively at gauge height wind speeds of 5 m $s^{-1}$ (Korchendorfer et al., 2017). This undercatch represents a large error in precipitation measurement, especially in cold regions, and could have a significant impact on water resource forecasting, climate trend analysis, and hydrological model initialization and validation (Barnett

et al., 2005; Pomeroy et al., 2007).

During the first WMO Solid Precipitation Intercomparison (1986-1993), the WMO recommended that it was necessary to designate a reference standard precipitation gauge for which all other precipitation gauges can be compared (Yang, 2014). The WMO recommended that a Double Fence Intercomparison

Reference (DFIR) be accepted as the standard for the measurement of solid precipitation (Goodison et al., 1998). The original recommended DFIR configuration consisted of a large (12 m), octagonal double fence, with a manually observed Tretyakov precipitation gauge in the centre. The decision to use the DFIR as a reference was based on intercomparisons with a Tretyakov "bush" shielded gauge at the Valdai experimental site where the DFIR closely matched the precipitation totals recorded by the bush shielded

gauge, which was considered to be a true estimate of snowfall (Golubev, 1986). A more recent long term (1991-2010), intercomparison between the DFIR and Valdai bush gauge by Yang (2014), documented that the Valdai bush gauge can measure up to 20-50% more snow over a 12 hour period than the DFIR for wind speeds of 6-7 $m\ s^{-1}$. This discrepancy in undercatch varies with wind speed, and improves during periods of lower speeds with undercatch of only 4-10% when speeds are less than 7 $m\ s^{-1}$ (Yang,

2014). The results of Yang (2014), clearly indicate that it is still necessary to correct the DFIR for wind induced undercatch of solid precipitation in order be used as a reference for true precipitation. For SPICE, the manual gauge inside the wind fence was replaced by an automated gauge and the configuration was called the Double Fence Automated Reference or DFAR (Nitu et al., 2012). The DFAR as a reference is traceable back to the bush shielded gauge at Valdai (Nitu and Roulet, 2016) but there really isn't an

independent verification of what the "true" precipitation amounts really are. This study presents a novel approach for using measurements of groundwater pressure in deep observation wells as an indirect

method for recording precipitation events. The ground water pressure measurements have the advantage of integrating over a much larger~~wider~~ area than traditional point location precipitation gauges (hectares vs a point measurement~~m²~~) and are not subject to wind-induced errors, providing an independent and potentially more robust measure of the true precipitation reaching the ground than a

DFAR/DFIR.

A method of measuring an uninterrupted record of the total moisture balance on a scale of hectares, utilizing measurements of groundwater pressures in underlying saturated formations, has been previously discussed in the literature (van der Kamp and Maathuis, 1991; Bardsley and Campbell, 1994,

2007; van der Kamp and Schmidt, 2017). The resulting moisture balance data derived from these groundwater observations are similar to those obtained by conventional $m^2$ scale weighing lysimeters, but on a much larger scale and with no significant hydrologic disturbance of the site. This moisture balance measurement technique has previously been referred to as an aquifer lysimeter (Bardsley and Campbell 1994) or a piezometric lysimeter (Barr et al., 2000), but more commonly has been called a geological

weighing lysimeter (Sopocleous et al., 2006; Bardsley and Campbell, 2007), and hereafter will be described as a geolysimeter. The geolysimeter has been described for measuring site water balance (van der Kamp and Schmidt, 1997; Barr et al., 200~~1~~0; Anochikwa et al., 2012), for evaluating hydrologic models (Marin et al., 2010) and for comparison with regional gravity changes as measured by the GRACE gravity satellite (Lambert et al., 2013). Previous publications have suggested that the geolysimeter method could

be used for inferring precipitation on a scale of hectares (van der Kamp and Schmidt, 1997; Barr et al., 200~~1998~~0, Sophocleous et al., 2006, van der Kamp and Schmidt, 2017), making use of piezometer data measured within low-permeability aquitards at depths of a few tens of meters. Bardsley and Campbell (1994), state that a case could be made that this technique is a better recorder than a rain gauge for brief intense precipitation events, because it has the advantage of integrating rainfall over a much larger ~~wider~~

area. The larger integration area for measuring precipitation would make this measurement method less susceptible to outlier measurement errors at point locations. Both Bardsley and Campbell (2007) and Barr et al. (2001) report a close correspondence between co-located geolysimeter and precipitation gauge measurements but do not include detailed quantitative analysis of this correspondence. ~~most of the previous studies have been more qualitative than quantitative, often due to the spatial separation of the~~

~~geolysimeter and the measuring precipitation gauges. Most p~~Previous intercomparisons also do not include a discussion on measurement of snowfall.~~However, the precipitation comparisons done so far~~

have been more qualitative than quantitative due to the spatial separation of the geolysimeter and the measuring precipitation gauges.

The objective of this paper is to analyze changes in the water-level records from a deep well geolysimeter and compare those to event-based precipitation records measured with a co-located precipitation gauge. This intercomparison will help to evaluate the potential use of geolysimeters as an independent and accurate reference measure of precipitation to be used for validating a variety of precipitation gauges, with a focus on providing an improved means of validating the measurement of solid precipitation.

## 2  Groundwater theory

The operating principle of a geolysimeter is that changes of total mechanical load above a deep confined geological formation are transmitted instantaneously to the groundwater pressure inside that formation. This loadstress transmission is a basic principle of soil mechanics. It has long been recognized in the groundwater literature, especially with respect to the analysis of the effects of atmospheric pressure changes on the water levels in deep observation wells (Jacob, 1940).

In the hydrogeology context, a "confined" formation is a saturated porous formation that is isolated from the shallow water table by overlying low-permeability formations. Changes in the water table elevation are at most transmitted only very slowly to the groundwater pressure in confined formations, and vice versa, changes of groundwater pressure in the confined formation are dissipated at most only very slowly by flow to the water table. For typical precipitation events, with duration of at most a few days, the induced groundwater pressure changes in a confined formations are not significantly dissipated by flow of the groundwater (e.g. van der Kamp and Maathuis, 1991; Anochikwa et al., 2012; Freeze and Cherry, 1979, p. 229; van der Kamp and Schmidt, 2017).

Changes of atmospheric pressure are a particular type of surface load that are easily measured with barometers allowing for a correction of these effects. Once the atmospheric effects have been removed, the responses of deep observation well pressure measurements to other types of surface load changes are observable. The change in surface load is born in part by the groundwater in the saturated pores of a

confined formation and in part by the solid skeleton of the formation. The proportion of the load change that is carried by the pore water is referred to as the "loading efficiency" and is constant for a particular observation site, being a property of the porosity and compressibility of the formation and of the pore water. Thus, the loading efficiency of a confined formation can be determined from its measured

response to atmospheric pressure fluctuations that are also recorded at or near the site of an observation well (e.g. Anochikwa et al. 2012). Typical values of loading efficiency are in the range of 0.60 to 0.95 for sands and 0.90 to 0.99 for clays and clay-rich glacial tills. The groundwater pressure in confined formations may also be subject to small earth tides, typically with magnitude of a few mm to a few cm in terms of water-level change. The earth tide effects can be removed by using the Tsoft code (van Camp

and Vauterin, 2005) to calculate the tidal acceleration at the location of the observation well (Anochikwa et al. 2012).

During a precipitation event on unfrozen ground, the water that falls on the ground either enters into the soil by infiltration or it runs off over the surface if the infiltration capacity of the soil is exceeded. Some

evaporation may also occur but it is generally small because the air near the ground tends to be near saturation during precipitation. For snow events on frozen ground, the snow accumulates on the ground surface with negligible infiltration or surface runoff, but wind and sublimation can result in the loss and/or re-distribution of snow in the tree canopies and on the ground. If losses of the precipitated water from the response area of a deep observation well by evaporation, sublimation, runoff and wind are very small,

then the total change of water load on the surface is equal to the precipitation that fell. This change of load can be accurately measured by means of measuring the pore water pressure inside deep observation wells.

A geolysimeter senses approximately 90% of the changes of total surface loading over a response area

with a radius of approximately 10 times the depth of the observation well if the geolysimeter is installed in an aquitard formation with low permeability (van der Kamp and Schmidt, 1997). For such formations, spreading out of the moisture loading signal by lateral flow is limited. For geolysimeters installed in permeable aquifers, as is the case for most observation wells, the responsesensing area may be larger if the moisture loading event is of long duration so that lateral groundwater flow in the aquifer distributes

the pore pressure changes resulting from the moisture load. For short-term events lasting at most a few days, such as individual precipitation events, the response area is likely to be quite well-defined by the

"radius equal to 10 times the depth" rule of thumb, in analogy with the limited spatial extent of groundwater pressure drawdown due to pumping for pumping tests which typically have durations of hours or a few days at most (Kruseman and de Ridder, 1994).

## 3  Experimental site

The experimental site is located in the northern prairie region of North America, 10 km north of the town of Duck Lake, Saskatchewan (Fig. 1a) at ~~13U  417810E,  5863437N~~ 52.92°N  106.22°W. The co-located observation well and precipitation gauge are located in an abandoned school yard, surrounded by a shelter-belt of pine and spruce trees (Fig. 1b). The regional setting and landscape of the Duck Lake observation well site is described by Marin et al. (2010). The well is completed at 124.6 m depth near the bottom of a deep confined aquifer which is overlain by 35 m of surficial sand and by an 82 m thick glacial till unit with extremely low bulk hydraulic conductivity of about $1\times10^{-11}$ m s$^{-1}$~~m/sec~~, or about 0.3 mm yr$^{-1}$~~mm/year~~. The area is partially wooded, gently undulating with a few metres of local relief, and very slight regional slope. A small fen about 200 m north of the site lies at the head of a shallow swale that conducts surface runoff during very wet conditions and which is defined as MacFarlane Creek further downstream. The winter-time base flow in the creek was estimated by Marin et al. (2010) to be about 40 to 80 mm yr$^{-1}$~~mm/year~~ on a watershed runoff basis, or about 0.1 to 0.2 mm /day$^{-1}$.

The well was instrumented with automated recording equipment in 2007 and a meteorological station in 2010. Prior to automation, water levels were recorded from 1964 onward with float-actuated chart recorders. The long-term record for the observation well (Duck Lake No. 2), plotted as monthly median values, can be found on the Saskatchewan Water Security Agency website, together with detailed information on the well completion data [www.wsask.ca/Water-Info/Ground-Water/Observation-Wells/] (last accessed Feb. 2017).

The long-term water level record for the deep well has been shown to reflect the total changes of water storage in the surrounding landscape on a month-by-month basis (van der Kamp and Maathuis, 1991; Marin et al., 2010), with well water level changes closely correlated with precipitation events, evapotranspiration and losses of water by surface run-off (van der Kamp and Schmidt, 2017). These previous studies demonstrate how the well acts as a large-scale weighing lysimeter. Water level records

for the Duck Lake well plus three other similar wells in southern Saskatchewan were compared to the GRACE gravity satellite changes in the region (Lambert et al., 2013) and showed correspondence between the multi-year water storage changes reflected in the well records and the regional change of mass as measured by GRACE.

The automatic meteorological station at the Duck Lake geolysimeter site measured air temperature and humidity at 1.5 m, wind speed and direction at 2 m, accumulated precipitation via a single Alter-shielded Geonor T-200B accumulating precipitation gauge, and snow depth. Observations were made and recorded every 30 minutes and are available from November 2010 through March 2016 and beyond

although with some breaks in the records, notably for most of 2012, due to various equipment failures.

## 4  Methods

The raw 30 min deep well observations, sampled at the beginning of each 30 minute period, require an adjustment for the effects of atmospheric pressure and earth tides in order to be comparable to

precipitation loading. Atmospheric pressure is measured by a pressure logger suspended inside the well casing above the water level. The process of adjusting for atmospheric pressure and earth tides is described in more detail by van der Kamp and Schmidt (2017). For these well observations, the loading efficiency was determined to be 0.798 on the basis of the observed response of the well to barometric pressure changes. The barometric pressure changes, multiplied by 0.798, were subtracted from the

recorded water-level changes. — and earth tide effects were removed using the TSoft code of van Camp and Vauterin (2005)The earth tide effects were removed using the Tsoft code — (van Camp and Vauterin, 2005) to calculate the tidal acceleration at the site (nm$^2$ /sec$^{-1}$), and subtracting these from the barometrically corrected water-level changes, — multiplied by a factor of 2.8x10$^{-6}$ determined by trial and error for optimal removal. The water-level changes with barometric and earth tide effects removed, were

multiplied by 1/0.798 = 1.253 to convert the short-term changes to moisture loading responses. The 30 minute well data was then smoothed using a Savitzky-Golay filter (Savitzky and Golay, 1964) and the positive increases accumulated for each event for comparison with the gauge observed precipitation.

The Geonor T-200B precipitation gauge had one vibrating wire transducer for the derivation of bucket

weight measurements. Precipitation falling through a 200 cm$^2$ orifice is collected in the bucket and

**Formatted:** Superscript

**Formatted:** Superscript

weighed by converting the observed frequency of the vibrating wire transducer to the corresponding bucket weight (in mm). The differential bucket weight over the measurement period becomes the total precipitation amount for that period. The sensor calibration was checked periodically by adding a known mass of water which was then compared to the sensor determined change in bucket weight to confirm near-zero calibration drift. The 30 minute bucket weight measurements were quality controlled by removing spurious and service related jumps. Prior to calculating 30 min precipitation amounts, the bucket weight time series was filtered using a "brute-force" technique of balancing positive and negative noise in the signal (Pan et al., 2016) until the accumulated positive changes exceed a threshold of 0.05 mm. The result is a smoothed time series of 30 minute bucket weights from which precipitation is calculated as a differential.

For the purpose of comparing the gauge precipitation to the change in well pressure, precipitation was aggregated to events, where an event is defined as a continuous precipitation period delineated by a break in precipitation greater than 3 hours. The main justification for aggregation is to allow enough snow to accumulate on the surface to solicit a response from the observation well. Rain events were aggregated in the same way for consistency. Precipitation events were categorized as either snow or rain by using 1.5 m air temperature. When the maximum air temperature for an event was less than -2° C, the event was categorized as snow. When the minimum air temperature for the event was more than 2° C, the event was categorized as rain. No mixed events were considered in this analysis and events smaller than 0.5 mm were removed to further decrease potential noise and erroneous precipitation measurements.

## 5  Results

Figure 2 illustrates typical responses of groundwater pressure in a deep observation well to precipitation events plotted together with the corresponding raw Geonor gauge record for both a rainfall (Fig. 2a) and a snowfall (Fig. 2b) precipitation event. The figure, along with the precipitation data which has been filtered, also shows both the raw well load chance (solid blue line) and the filtered and accumulated well load change (blue dashed line).

The rain event for June 4-5, 2010 (Fig. 2a) of about 20 mm shows the response of the moisture loading signal in the well to the accumulating rain event. However, the moisture load change is clearly smaller than the gauged precipitation by about 24 mm (using the accumulated load change). The likely reason for the discrepancy is water loss from the area by surface outflow as indicated by the sharp decline of moisture load during the night-time hours immediately after that event and in the continuing decline in the following days. Evapotranspiration was likely very small since relative humidity during the night and following the precipitation event was 100 %., although tThe decline in water level in the well from 18:00 UTC (12:00 local) to 4:00 UTC (22:00 local) prior to the event is likely indicative of evapotranspiration.Evapotranspiration was likely minimal since relative humidity during the night during and following the precipitation event was 100 %. Since the summer of 2010 was unusually wet in this region, with flooding reported in many places, it is likely that fens near the study site became hydrologically connected resulting in a net water loss fromwithin the response area of the well, which lies in the headwater area of MacFarlane Creek.

The winter snow event of March 10-11, 2011, illustrated in Fig. 2b, shows a close correspondence between the gauged cumulative precipitation and the moisture load change. At this time of the year, evapotranspiration and runoff were not substantialignificant. The temperature varied in the range of -3° to -16° C and wind speed varied between 0.5 and 1.1 m /sec$^{-1}$, indicating that wind induced gauge bias and snow re-distribution and sublimation were minimal.

During the observation period between 2010 and 2016, a total of 103 events (56 snow and 47 rain events) were observed varying in length from 9 to 108 hours. The mean event length for snow was 38 hours while the mean event length for rain was 46 hours, although these event lengths are artificially increased by several hours both at the beginning and the end of the event to provide a good baseline for both gauge and well observations.

Summary statistics for the comparison between the geolysimeter and the gauge event precipitation comparison are shown in Table 1 and the scatter plot and regression lines are shown in Fig. 3. The correlation coefficient, $r^2$, is 0.99 for both the combined rain and snow precipitation events (All) and for rainfall events. The correlation is 0.94 for snow events. The slopes of the regression line are consistent at 0.90 for All and Rrain and 0.93 for Ssnow. RMSD (Root Mean Square Deviation)E varies from 2.3 mm for

rain (with a total gauge rainfall of 903 mm) to 0.8 for snow (with a total gauge snowfall of 224 mm). With the rain and snow events combined, the geolysimeter shows a negative bias of 35 mm. For rain, the geolysimeter has a negative bias of 59 mm which is illustrated by the rain regression line (black dotted line with open circles) shown in Fig. 3. Figure 3 illustrates the increasing degree of geolysimeter underestimation for rainfall~~precipitation~~ events of a larger magnitude. This can likely be explained by the larger percentage of surface run-off that occurs during large rainfall events versus smaller events. The magnitude of the bias for rain appears to be related to the total event amount (Fig. 4) with an $r^2$ of 0.51. For snow, the bias is positive at 23 mm, with no relation to the total precipitation amount ($r^2 < 0.10$). The comparison for snow, rescaled in Fig. 5 (blue dashed regression line), shows the consistent geolysimeter overestimation (or the precipitation gauge underestimation) of snowfall events of all magnitudes.

Given the propensity for precipitation gauges to underestimate snowfall, we suspect that it is more likely that the precipitation gauge is underestimating rather than the geolysimeter overestimating snowfall. Although Fig. 6 suggests that almost 65% of the 30-minute periods during precipitation events have gauge height wind speeds less than 1.75 m /s$^{-1}$, wind speeds can exceed 3 m s$^{-1}$~~m/s~~ on occasion. For this reason, we chose to adjust the 30-minute precipitation amounts for wind induced errors using the transfer functions described by Kochendorfer et al. (2016, 2017), developed using a sub-set of the WMO-SPICE dataset. The Kochendorfer (2016) paper presents two transfer functions, a complex sigmoidal transfer function originally presented by Wolff et al. (2015) and a simpler exponential-arctan transfer function. The exponential-arctan coefficients were later revised by Kochendorfer et al. (2017) using data from 8 more— SPICE sites representative of varying climatic conditions, making the transfer function potentially more broadly applicable. For the sigmoidal function, we use the coefficients presented in the earlier paper (which were fit to the data from Marshall, CO, USA) and the more recent and broadly applicable coefficients for the exponential-arctan function. Of the 56 total snowfall events, only 51 could be adjusted due to missing wind speed or temperature. This explains the difference in the "Unadjusted" statistics shown in Tables 1 and 2. Each 30-minute period within each event was adjusted individually using the average wind speed and temperature during that 30-minute period. As before, the high frequency data is then accumulated for each of the snowfall events. As shown in Fig. 6, winds speeds were generally low such that most adjustments are minor. This is reflected in Fig. 7 that shows the unadjusted snowfall (blue dots), the sigmoidal adjustment (red squares), and the exponential-arctan adjustment (black boxes) compared to the geolysimeter. The larger adjustments are evident when the

markers representing the adjusted precipitation are shifted farther to the right in the figure. Larger adjustments were common during precipitation events of a larger magnitude (presumably, also of longer duration). Application of the two transfer functions decreased the total bias of the gauge as compared to the geolysimeter with the sigmoidal and exp-arctan functions reducing the bias from 10.2% to 7.5% and 3.1% respectively. However, the RMSD̶E̶ was not reduced very much by the adjustment with only a slight decrease from 0.86 mm to 0.85 mm for the sigmoidal adjustment and an increase to 0.93 mm for the exponential-arctan adjustment. These summary statistics for the adjusted events are shown in Table 2.

## 6  Discussion

Detailed inspection of the moisture loading record provides a strong indication that net runoff out of the response̶s̶e̶n̶s̶i̶n̶g̶ area of the geolysimeter occurred during some of the more intense rain events (e.g. Fig. 2a̶b̶) which is reflected by the increase in geolysimeter bias with increased rainfall (Fig. 4). Since hydrological dynamics are complex, the occurrence of runoff is not always directly correlated with increased rainfall amount. Evapotranspiration, especially from the tree canopies, may have also resulted in some moisture losses during the rain events, especially the events of longer duration (which are often related to total rainfall amount). Although we did not attempt to estimate evapotranspiration, we do see instances where the relative humidity measured during some events dropped below 100 %, indicating the potential for some evapotranspiration.

Some other considerations that have the potential to impact the timing and magnitude of the p̶e̶a̶k̶ geolysimeter precipitation estimates as shown in Fig. 2 are the temporal resolution of the geolysimeter observations and the data filtering process. The effect of observation resolution is possible because the response of the geolysimeter to rainfall loading is nearly instantaneous meaning that the minimum or the peak water level in the well may have been missed by the water level readings that were taken once every 30 minutes.o̶c̶c̶u̶r̶r̶e̶d̶ ̶f̶o̶l̶l̶o̶w̶i̶n̶g̶ ̶t̶h̶e̶ ̶p̶e̶a̶k̶ ̶3̶0̶-̶m̶i̶n̶u̶t̶e̶ ̶o̶b̶s̶e̶r̶v̶a̶t̶i̶o̶n̶ ̶s̶h̶o̶w̶n̶ ̶i̶n̶ ̶t̶h̶e̶ ̶f̶i̶g̶u̶r̶e̶,̶ This may result in an under̶l̶o̶w̶e̶r̶ estimate of precipitation. This effect would only be significant if water losses from the geolysimeter response area by runoff or evapotranspiration were significant during the 30 minutes before the beginning or after the end of the precipitation event. Considering the low relief of the study area, runoff is slow (cf Fig. 2a) and the error due to the sampling interval is likely to be much smaller than 1 mm.̶ The impact of the data filtering process may be more substantial in summer. The Savitzky-Golay

filter de-spikes and smooths the data which tends to have some inherent noise (cf Fig. 2). When precipitation is intense, as in Fig. 2a, the filter tends to underestimate the well response and could explain some of the bias in the geolysimeter during more intense convective events. As Pan et al. (2016) suggest for precipitation data, the filtering technique has the potential to impact results, and this can also be said for the geolysimeter. More work is needed on this topic for processing data from both sources.

The rainfall intercomparison also does not account for spatial scaling of precipitation, especially convective precipitation, when comparing the point gauge measurement to the more spatially distributed geolysimeter measurement. Highly localized rainfall which is a characteristic of summer convection may not be uniform across the geolysimeter response area, perhaps resulting in the geolysimeter under-reporting precipitation as compared to the gauge. Studies such as De Michele et al. (2001) suggest the use of an areal reduction factor (ARF) to scale point measurements to spatial estimates and a rough approximation for an ARF of 95% would more closely align the gauge and the geolysimeter and could explain much of the bias. However, ARFs for the general location and climatology of this field site are not well-known or understood. This is complicated further by the geolysimeter measurement principle where the response of the geolysimeter, located in the centre of the response area, has reduced sensitivity to load changes per unit area with distance from the centre.

If we make the assumption that the underestimation of rainfall by the geolysimeter is a result of evapotranspiration and runoff during a rainfall event and that these processes are negligible during snowfall events, then there is potential for using geolysimeter measurements of snowfall as an independent reference for the measurement of solid precipitation. However, the landscape and surface characteristics of the geolysimeter response area must also be considered such that wind redistribution and sublimation are minimized (i.e. the area has adequate snow catchment and retention properties, such as vegetation cover, and reduced environmental exposure to wind). If these criteria are met, then in theory, this would allow for an independent measure of solid precipitation for developing and validating transfer functions used to adjust undercatch of the gauge measurement of snowfall.

The application of the two transfer functions presented by Kochendorfer et al. (2016, 2017) both result in an improvement in the total bias of the gauge measurements as compared to the geolysimeter with the simpler exponential-arctan function representing the greatest improvement in the bias. However, neither

transfer function improves the RMSD~E~, which is consistent with what Kochendorfer et al. (2017) showed with testing on the SPICE data. Because of the low wind speeds at the Duck Lake site as represented in Fig. 6, this really is not a robust test of these transfer functions, as the total adjustment is very small. It is proposed that a supplementary intercomparison site be installed outside of the sheltered area within which ~~that~~ the precipitation gauge is currently installed ~~in~~ but still well within ~~very close to~~ the 1 km footprint radius of the geolysimeter. Gauges outside of the sheltered area would be exposed to more typical windy conditions found during snowfall on the Canadian Prairies. With wind speeds during precipitation averaging close to 5 $m\ s^{-1}$~~m/s~~ and often exceeding 10 $m\ s^{-1}$~~m/s~~, the transfer functions derived from WMO-SPICE could be more thoroughly tested against the independent geolysimeter measurements. Further work is also required to determine the minimum temporal resolution of the geolysimeter, especially during (light) snowfall events.

Both methods of measuring precipitation, whether via conventional gauges or via a geolysimeter, have their limitations. The wind bias in the gauge measurement of snowfall is well documented. Gauge measurements can also be fraught with other issues such as capping (the plugging of the orifice with accumulating snow), poor or infrequent maintenance (resulting in overflowing, bucket freezing, etc.), and mechanical failure. The gauge measurement is also just a point measurement and may or may not be spatially representative. Although the geolysimeter does not suffer from many of the same issues as the gauge measurement and is more of a spatial estimate of precipitation, it also has its limitations. The technique cannot be used everywhere due to geologic requirements (i.e. the aquifer needs to be confined and not impacted by human activity). Also, cumulative time series of precipitation are more difficult to produce with a geolysimeter since the long term record can be impacted by slow ground water storage changes (e.g. seepage, ponding, pumping, etc) which are more difficult to compensate. In winter, the response area of the geolysimeter cannot be a region of localized accumulation (i.e. from drifting snow) or scouring, so redistribution in the response area needs to be a random process. For deep observation wells with response areas of several $km^2$, redistribution of snow within the response area can be assumed to be a random process as long as the landscape is relatively homogenous. However, the sublimation of blowing snow at exposed sites, even if the redistribution of snow is random, could result in underestimates of snowfall during longer events. Given the limitations of both techniques, the geolysimeter could certainly help compliment and improve conventional precipitation measurements where geological and landscape conditions are favourable for their co-location.

## 7 Conclusions

This study shows that it is possible to make an accurate estimate of event based precipitation using a deep well geolysimeter. Although the geolysimeter underestimates rainfall by 7% and appears to overestimate (unadjusted) snowfall by 9%, the correlations are high with an $r^2$ of 0.99 and 0.94 for rain and snow respectively. The underestimation of rainfall, especially for larger events, can be linked to the net loss of water from the geolysimeter responseintegration area during the events. The exact mechanisms associated with this net loss have not been documented here, but they are likely related to both run-off and evapotranspiration and would necessitate the installation of additional instrumentation to more accurately quantify. However, assuming that runoff and evapotranspiration/sublimation of snow on the ground are negligible during winter snowfall events, and that net wind re-distribution of snow out of the response area is minimal, the accuracy of the geolysimeter should be high for measuring snowfall at this location. At least some of the apparent overestimation of snowfall by the geolysimeter is likely due to the undercatch of snowfall by the precipitation gauge. Although wind speeds in the sheltered area are relatively low and therefore the bias adjustments are small, the applied bias adjustments result in a reduction in the difference between the geolysimeter and the gauge from 9% to 3%. Based on these results, the geolysimeter can be used as an accurate independent reference measurement of solid precipitation and as a complement to conventional techniques for long term precipitation monitoring.

## Acknowledgements

The authors would like to thank the Saskatchewan Water Security Agency for their continued collaboration and access of the observation well located near Duck Lake SK. We would also like to express our gratitude to the reviewers who have provided their time to help us improve this manuscript.

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

**Formatted:** Font color: Red

# Tables

Table 1: Summary statistics for the comparison of geolysimeter estimated precipitation event amounts to corresponding gauge precipitation event amounts at the Duck Lake site. Gauge precipitation observations are considered to be the independent variable where β is the slope and ε is the intercept of the regression line. Bias is the gauge precipitation subtracted from or divided by the geolysimeter precipitation.

| Precipitation Type | $r^2$ | β | ε | RMSDE (mm) | Total Gauge Precip (mm) | Bias* (mm) | Bias* (%) |
|---|---|---|---|---|---|---|---|
| All | 0.99 | 0.90 | 0.79 | 1.7 | 1127 | -35 | -3.2 |
| Rain | 0.99 | 0.90 | 0.70 | 2.3 | 903 | -59 | -7.0 |
| Snow | 0.94 | 0.93 | 0.70 | 0.8 | 224 | 23 | 9.3 |

*positive value indicates that the gauge is measuring less precipitation than the geolysimeter

Table 2: Summary statistics for the comparison of the unadjusted and adjusted (sigmoidal and exp-arctan functions) gauge snowfall measurements with the geolysimeter. Bias is the total gauge precipitation subtracted from (mm) or divided by (%) the total geolysimeter precipitation. Note that the adjustment was performed only on 51 of the 56 snowfall events due to some missing meteorological data; hence the % bias is slightly different than that reported for all snowfall events in Table 1.

| Adjustment | $r^2$ | RMSDE (mm) | Bias* (mm) | Bias* (%) |
|---|---|---|---|---|
| Unadjusted | 0.95 | 0.86 | 24 | 10.2 |
| Sigmoidal | 0.95 | 0.85 | 18 | 7.5 |
| Exp Arctan | 0.94 | 0.93 | 7 | 3.1 |

*positive value indicates that the gauge is measuring less precipitation than the geolysimeter

## Figures

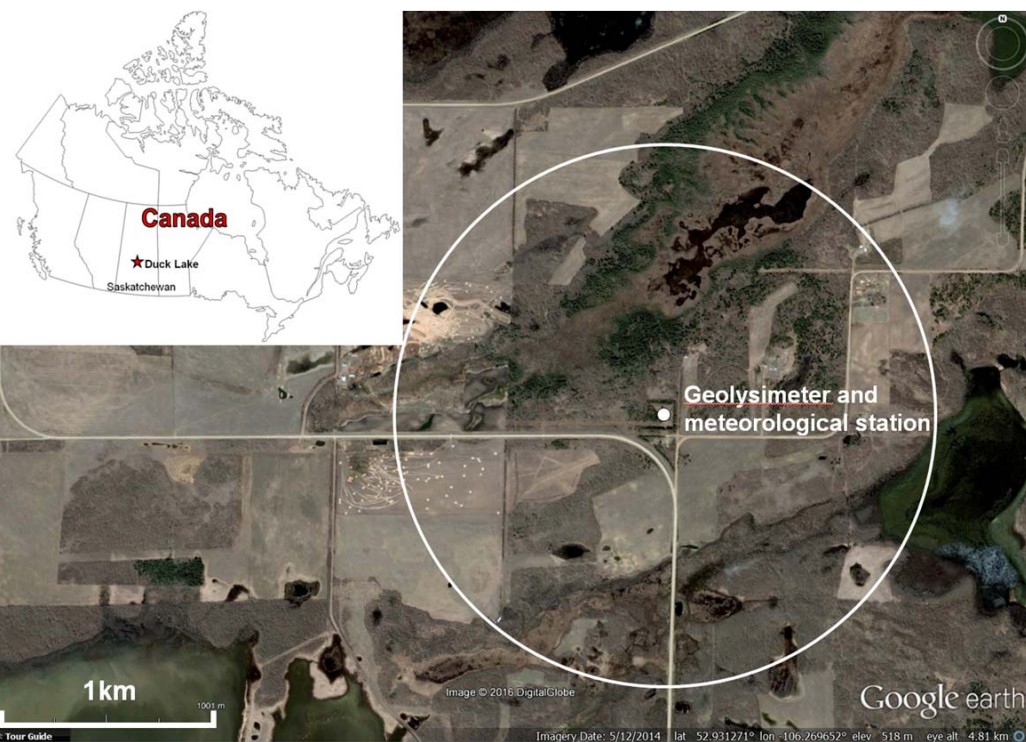

Figure 1: Location of the experimental site ~10 km north of Duck Lake, Saskatchewan, Canada (inset) and the location of Duck Lake No. 2 observation well and meteorological station centred in the geolysimeter response area with a radius of ~1.25 km or 10 x the well depth (white circle).

**a)**

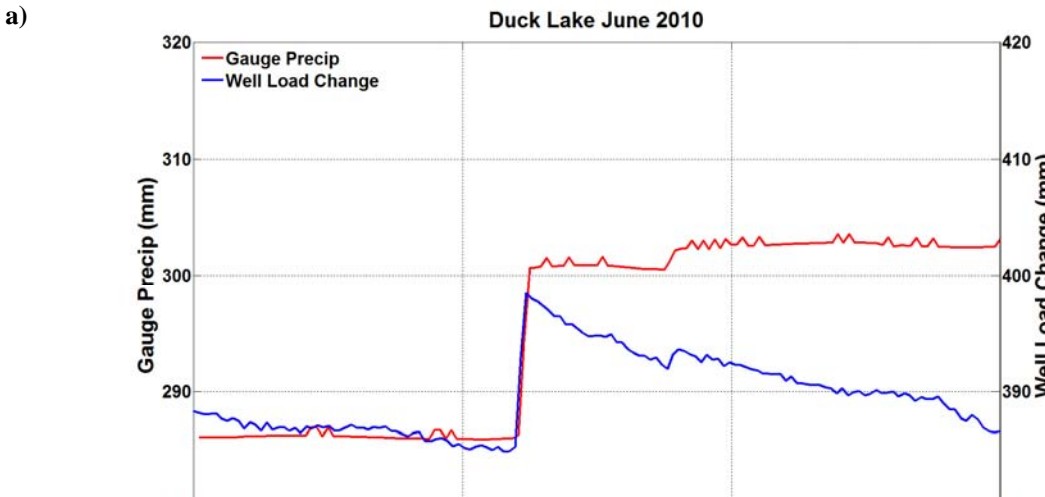

**b)**

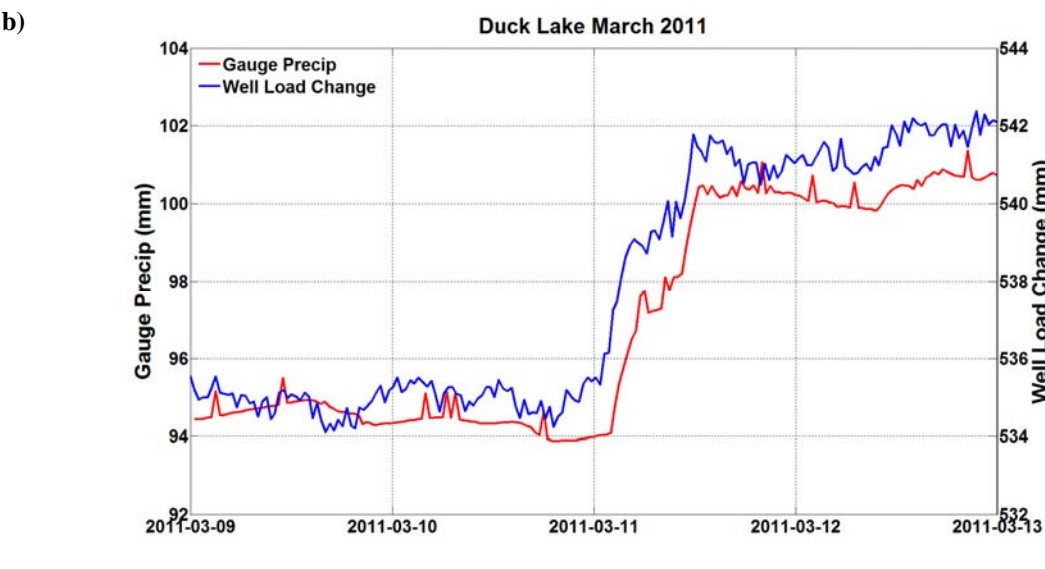

a)

b)

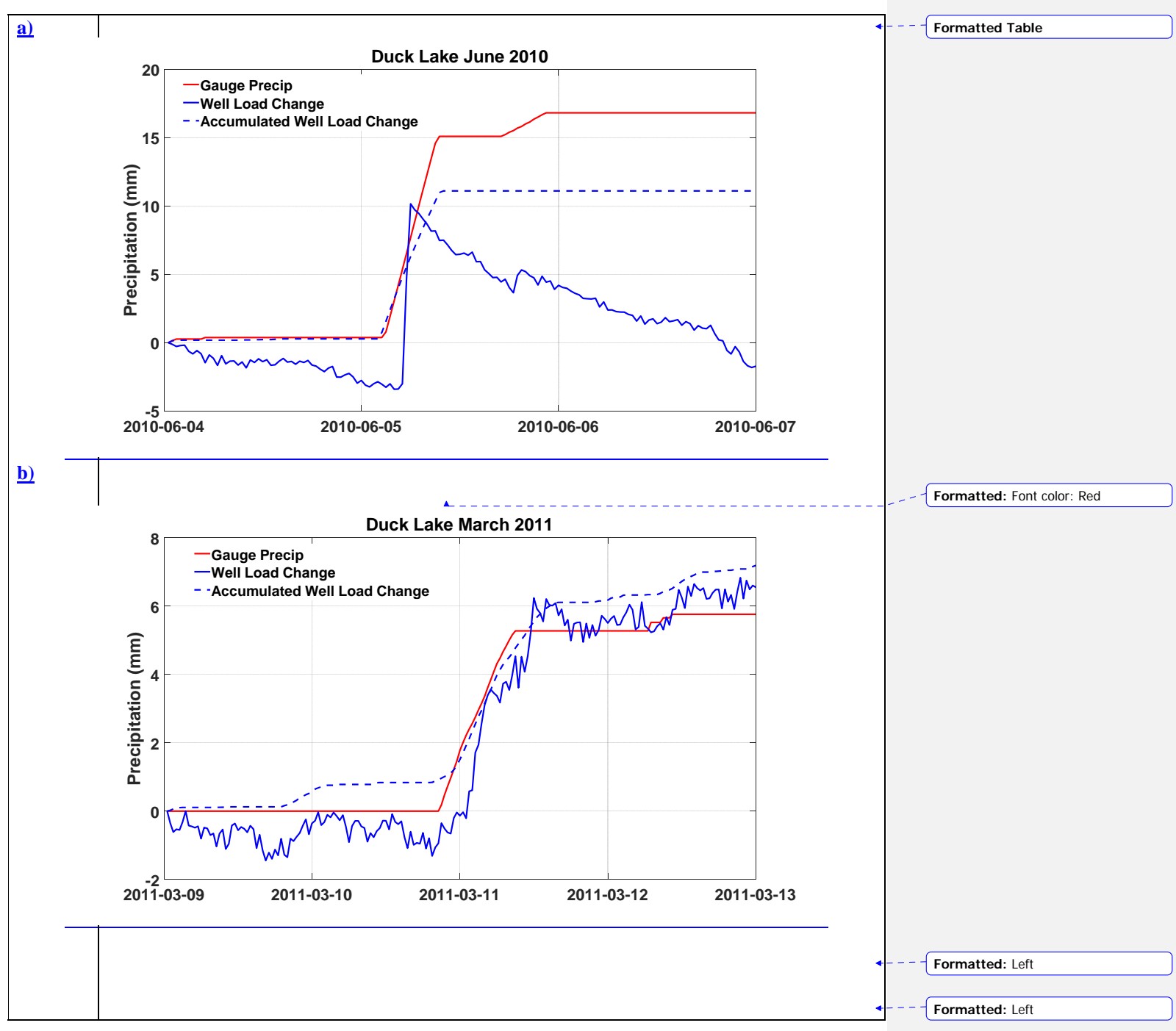

Figure 2: Water level record for Duck Lake No. 2 observation well (piezometer) compared with the ~~raw (unsmoothed and unfiltered, including signal noise)~~ accumulated precipitation from the gauge ~~bucket weight~~ at the site for a) June 5-6, 2010 rainfall event and b) March 11-12, 2011 snowfall event. The observation well record has been corrected for the effects of atmospheric pressure changes and earth tides and multiplied by 1.253 to convert the water-level change to an equivalent moisture load change. Time is GMT (LST + 6 hours). The precipitation data (red solid)has been filtered and accumulated while both unfiltered (blue solid) and filtered and accumulated (blue dashed) well data are shown.

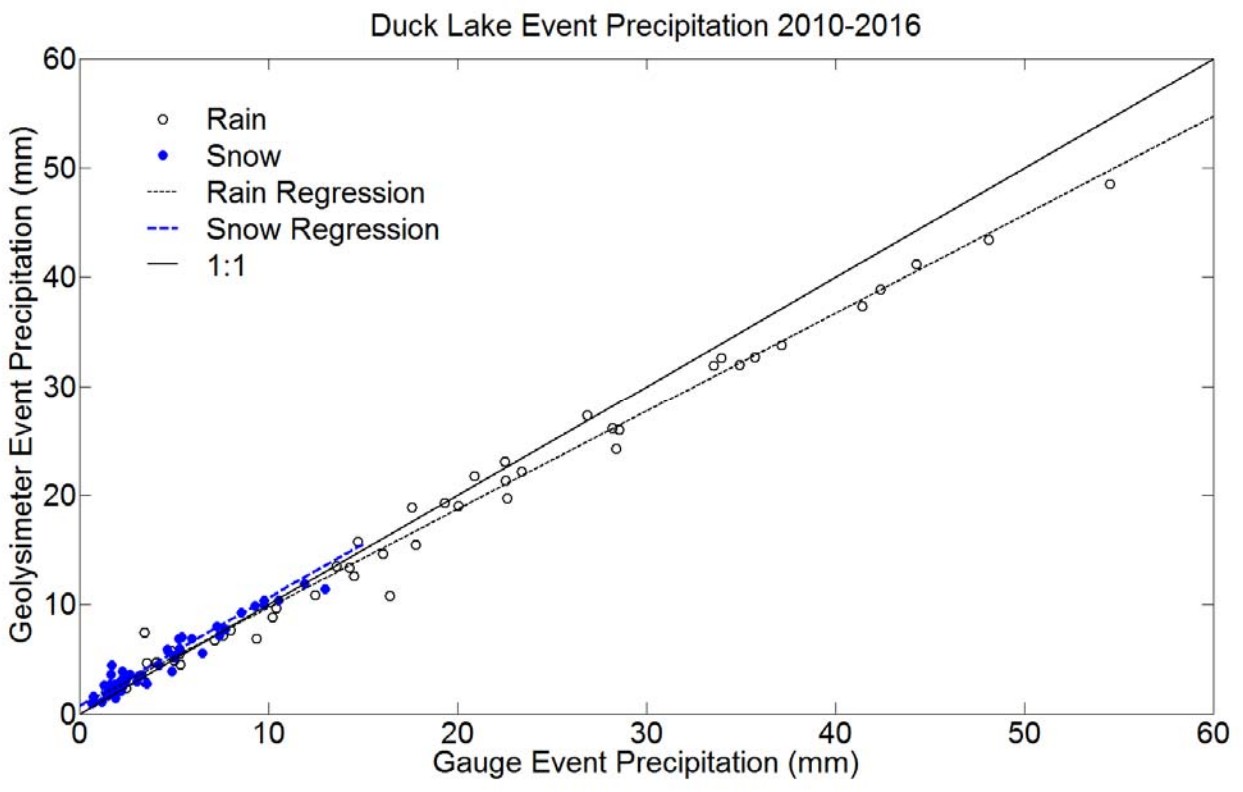

Figure 3: Duck Lake geolysimeter event precipitation compared with gauge event precipitation separated into rain and snow. Regression lines for rain and snow and the 1:1 line are also shown.

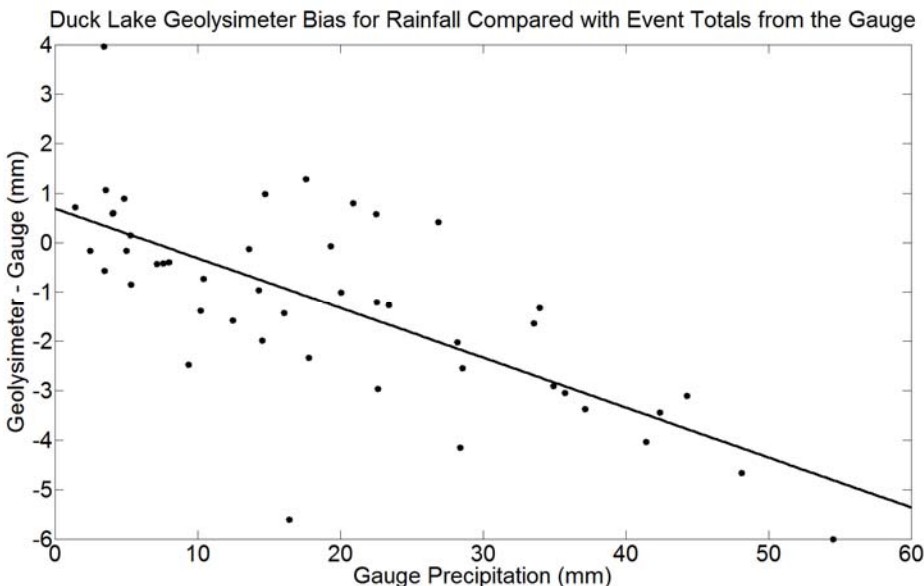

Figure 4: Relationship between the bias in geolysimeter rainfall measurements and total event rainfall amount as measured by the gauge at Duck Lake ($r^2$=0.51).

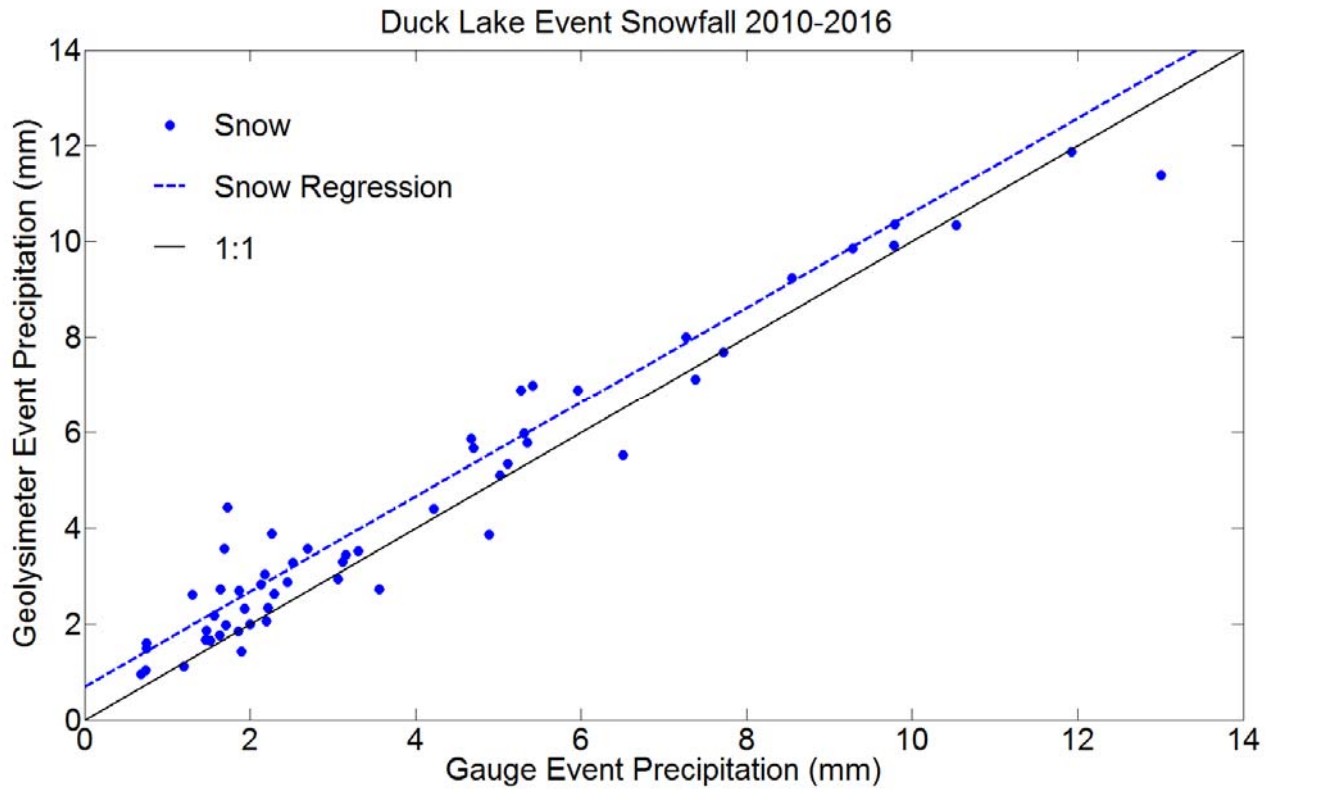

Figure 5: Duck Lake geolysimeter event snowfall compared with gauge event snowfall. Regression line for snow and the 1:1 line are also shown.

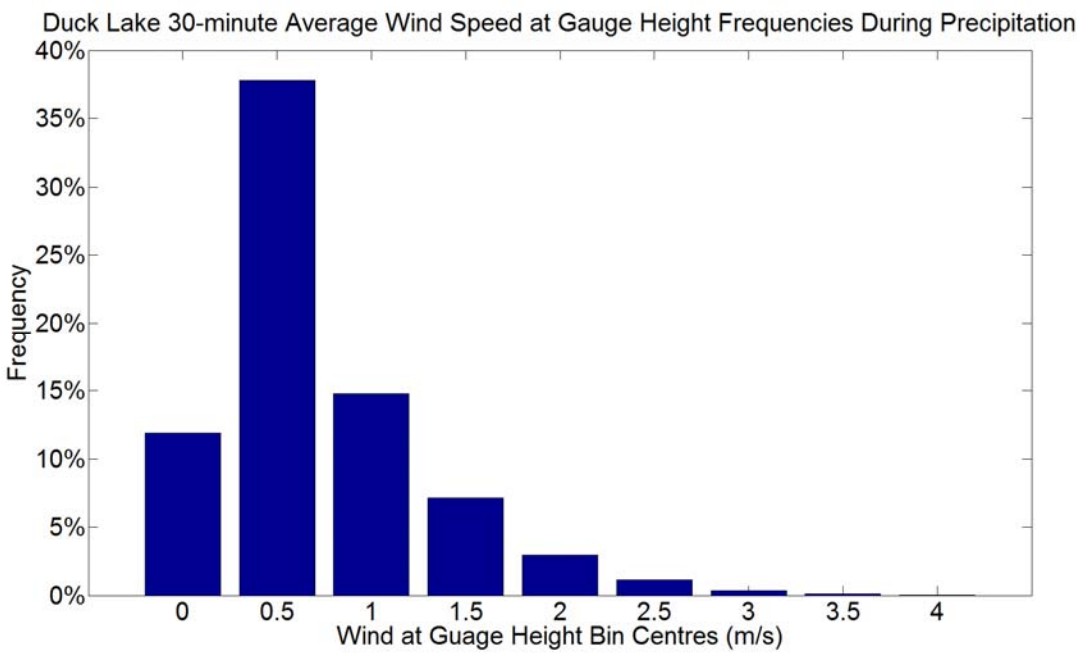

Figure 6: 30 minute average wind speed at gauge height frequency distribution during precipitation events at the Duck Lake geolysimeter site.

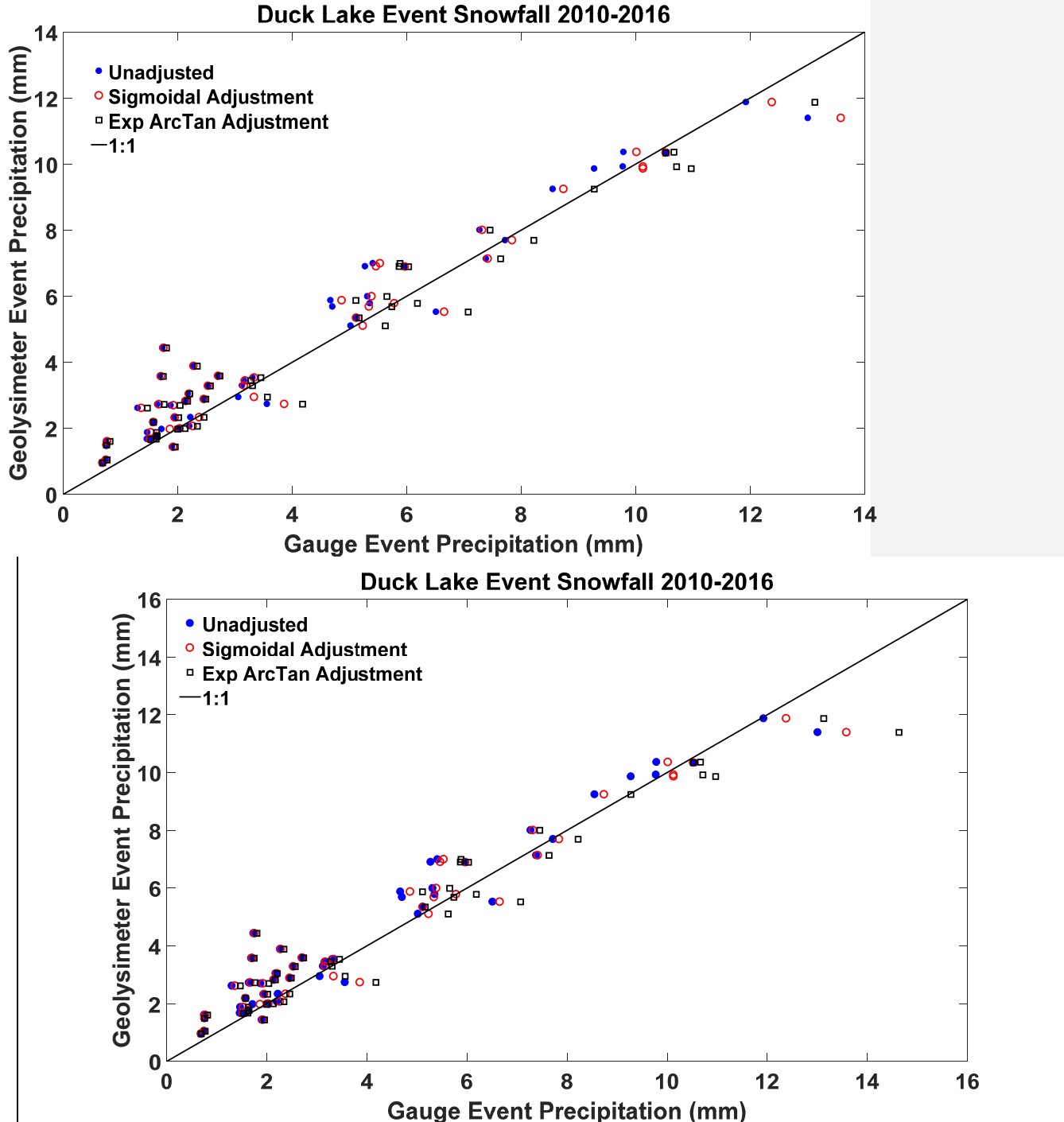

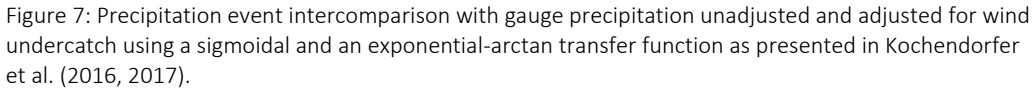

Figure 7: Precipitation event intercomparison with gauge precipitation unadjusted and adjusted for wind undercatch using a sigmoidal and an exponential-arctan transfer function as presented in Kochendorfer et al. (2016, 2017).