# Peer review of "Measuring precipitation with a geolysimeter"

_Hydrology and Earth System Sciences, 2017_

## Short Comment (SC1) · 28 Apr 2017

This paper seems a useful contribution to the geolysimeter literature. It's just a pity we can't get more geolysimeters set up around the world – perhaps as part of an International Continental Drilling Program initiative.

Just one clarification, however. It is stated in the paper:

"However, the precipitation comparisons done so far have been more qualitative than quantitative due to the spatial separation of the geolysimeter and the measuring precipitation gauges. "

The New Zealand geolysimeter ran for a number of years and in fact the rain gauge was located at the site for the specific purpose of recording precipitation observations that could be applied directly for pore pressure comparisons. Our data indicated that the site could serve at times as a giant rain gauge, with good matching of geolysimeter

observations to rainfall (Bardsley and Campbell, 2007).

Bardsley and Campbell, 2007. An expression for land surface water storage monitoring using a two-formation geological weighing lysimeter Journal of Hydrology, 335, 240-246.

---

## Author Comment (AC1) · 28 Apr 2017

Thank you for suggesting the Bardsley and Campbell reference. I will certainly have a look through the paper and add it to the list of references.

---

## Referee Comment (RC1) · E. Bardsley (Referee) · 21 May 2017

Comments on:

*Measuring precipitation with a geolysimeter*

By C.D. Smith et al.

General:
The paper is a useful contribution to the literature, in this case with the novelty aspect of using the area-integrating capabilities of geolysimeters as a snow measure to offset in part the high spatial variability of snow amount.

My comments are all of a minor nature related to the rainfall aspect, but might be taken into consideration in a revision.

Page 2 line 25

*However, the precipitation comparisons done so far have been more qualitative than quantitative due to the spatial separation of the geolysimeter and the measuring precipitation gauges.*

As noted in the independent comment, for the Bardsley-Campbell geolysimeter there was never at any time a spatial separation of the geolysimeter and the rain gauge. Is it really the case that in all other cited geolysimeter studies there was spatial separation between rain gauge and geolysimeter?

Top of page 7.

Were the 30 minute well water levels recordings of the well water levels every 30 minutes, or the average of higher time resolution monitoring from the previous 30 minutes?

Fig. 2

The slope change in water level following the rain event in Fig. 2a is certainly a good argument for site discharge loss being the cause of the evident slight under-estimation of the rain event. However, is this possibly a time resolution effect? That is, the pore water pressure increase from surface loading is instantaneous for practical purposes but there will be some finite time (a few minutes?) before the site rainwater starts to depart as discharge after a sudden event. What was the duration of the rainfall event in Fig. 2a? If it was less than 30 minutes and water levels were recorded every 30 minutes, is it possible for the maximum rise of water level to have been missed due to the relatively coarse sampling interval?
Some comment might also be added as to the likely effect (or not) of 1 km spatial variation of rainfall, given that precipitation is a point measure and the geolysimeter is a spatial average. It is a pity that at least one more precipitation gauge was not in operation at the site, but presumably other measurements of closely spaced gauges in similar environments might be mentioned in this respect.
Some comment should be made about the cause of the declining trend in water level prior to the rain event – evaporation and / or net groundwater export from the site?
The zero point of the rainfall plot should be set to correspond with the start of the rainfall event

The rainfall representation in both plots of Fig 2 is confusing and should be converted to cumulative rainfall (no negative slopes).

Earl Bardsley

---

## Referee Comment (RC2) · Anonymous Referee #2 · 23 May 2017

This paper could make a useful contribution by quantifying the relationships between a precipitation gauge and a geolysimeter. The authors have a done a good job of identifying many of the hydrological processes which can account for some of the differences between the sets of measurements, particularly those of the snowfalls.

Unfortunately, the authors have not adequately accounted for the difference between the areas of the rain gauge and the geolysimeter. The areal reduction factor, which quantifies the reduction of rainfall extremes over a region, compared to a point, is well known in hydrology. ARF values have been derived for many regions and are a standard part of engineering hydrology. Because the area of the geolysimeter is so large (almost 5 km$^2$) it approaches the sizes of the regions referenced in some published areal-reduction factor curves.

More theoretical analyses (De Michele et al, 2001, among others) also demonstrate that the reduction factor is related to the size of an event, which is also shown by the

plot of the geolysimeter and gauged rainfalls in Figure 3. However, the reduction factor also depends on the length of the event, while the authors have combined events of varying lengths. It would be possible to compare areal reduction factors for intensities, durations and frequencies derived from the data with published values.

At the very least, the effect of the area of the geolysimeter on the difference between its rainfall estimates and those of the gauge needs to be addressed.

De Michele, Carlo, Nathabandu T. Kottegoda, and Renzo Rosso. "The derivation of areal reduction factor of storm rainfall from its scaling properties." Water Resources Research 37, no. 12 (2001): 3247-3252

General The writing needs revision. The language is excessively colloquial and the terminology is frequently sloppy. A few examples are shown below

Page 1, Line 15 "Correlations varied from 0.99 for rainfall to 0.94 for snowfall." I believe that you are referring to the correlation coefficients of the linear regressions ($r^2$) rather than values of correlations between the data sets.

P 3, L 3 "wider area" Area is not the same thing as width! This sloppy usage is repeated throughout the document.

"(hectares vs $m^2$)" The exact areas of the gauge orifice and of the geolysimeter and their ratio should be given. This sentence grossly understates the ratio, i.e. the ratio of 1 hectare to 1 $m^2$ is 10,000:1. According to the manufacturer's website, the gauge orifice area is 200 $cm^2$, i.e. 0.02 $m^2$. If the radius of the geolysimeter measurement area is 1.25 km (as stated), then the ratio is more than 245 million to 1!

P 4, L 7 "This stress transmission" The previous sentence refers to the load (i.e. a force) and the pore-water pressure, not to a stress. Please make this clearer.

L 27 "at 13U 417810E, 5863437N." Why not specify the location by its longitude and latitude? They are global values, rather than being specific to a region, and are more easily understood.

P 7, L 2 "and earth tides"

Earth tides were not mentioned previously, when discussing the adjustment of the raw data, but probably should have been.

P 8, L 13 "Evapotranspiration was likely minimal since relative humidity during the night ..."

Does plant transpiration of water ever occur at night? The word "minimal" is being used in a colloquial sense. It would be better to say "very small".

L 21 "significant" This word should not be used in a scientific paper, unless you are giving the level of significance.

P 9, L 1 "Summary statistics ..." How were these computed? What program did you use?

L 5 "RMSE varies ..." The abbreviation should be defined. Also, since the gauge data are also believed to be in error, what you are actually computing is the root mean squared deviation (RMSD) between the two datasets.

P 10, L 29 What is an "adequate snow catchment"?

Figures P 18 Figure 1 caption "response area of ∼1.25 km" Area is not measured in km. This would appear to be the radius of the geolysimeter response area, correct?

P 24 Figure 7 It appears that a point is missing from the plot. There is a point plotted for largest gauge unadjusted precipitation, and for the sigmoidal adjusted value, but there does not appear to be a corresponding point for the exponential arctan adjusted value.

---

## Author Response (AR1)

**Response to the Editor Report and Reviewer's comments**

Thank You Mareile for acting as the handling editor for this manuscript and for the special issue. Not only will your assistance help us to improve this manuscript, your work with the special issue contributes substantially to the success of the SPICE project. We interject our response to your comments and the reviewer's comments below.

**Editor Report**

Editor Decision: Publish subject to revisions (further review by Editor and Referees) (08 Jul 2017) by Mareile Wolff Comments to the Author:

Dear authors,

Thank you for your submission to HESS and the inter-journal SI on The World Meteorological Organization Solid Precipitation InterComparison Experiment (WMO-SPICE) and its applications. Comparing the changes in the water-level records from a deep well geolysimeter to event based precipitation measurements of a co-located precipitation gauge connects two independent measurement methods from two closely related science branches: meteorology and hydrology.

To ease the understanding of the paper for both scientific communities, I suggest that you add some details about the principles of the different measurement methods. Especially the geolysimeter and its technical installation in general and more specific on your site would be helpful.

Much of the technical detail of the geolysimeter installation can be found in the references cited in the introduction. As you know, there are many areas of specialization in hydrology; ground water hydrology being an example. We are reluctant to get into too much technical detail beyond what is already discussed in the "Groundwater theory" section (Pages 4-5), anticipating that most readers either will not want to be too concerned with these details or if they are, will be able to find those details in the citations. I think it would be useful and interesting to include more detail about the site geology which makes the geolysimeter measurements possible, so as you suggest, we will add more detail here. We added a few sentences to technically describe the Geonor measurement in the Methods section on Page 7.

Please also describe the limitations of each measurement method, their differences and discuss how those differences were considered in your analysis. The two reviewers have pointed out some of these issues.

We are addressing these in more detail and have added a brief discussion of scaling issues as indicated in the response to Reviewer #2. The limitations of both methods are now discussed in a paragraph in the Discussion section.

Please comment on how applicable the described method may be for other places? What conditions needs to be in place to "extend" a meteorological site with a geolysimeter? The main problem for precipitation gauges is the undercatch of solid precipitation due to high wind speeds, typically at exposed sites. From your paper, I understand that also the area of a geolysimeter should be chosen with care to minimize wind redistribution and sublimation. Wind exposed sites may therefore also be problematic for a geolysimeter?

Those are good points. For measuring snow, it is important that the response area of the geolysimeter is neither an area that collects blowing snow nor is an area subjected to scouring. This shouldn't be a problem if the response area is large enough such that these processes are random and therefor are balanced. For an exposed site, the geolysimeter should be unaffected by redistribution provided that the flux of blowing snow onto and away from the response area is balanced. Sublimation is also a source of error in the winter which could be minimized by limiting event lengths but you are correct in assuming that these errors would be larger at exposed sites. Also, the groundwater geology has to be correct (i.e. existence of a confined aquifer) for the technique to work. We will address this more in the discussion.

You are now asked to submit a point-by-point reply addressing all the reviewers comments and suggestions, and prepare an upload of a revised manuscript accordingly.

In your revised version, please follow HESS' English guidelines and house standards; e.g. replace "m/s" with "ms-1" and mm/year" with "mm a-1".

Yes, thank you.

Yours sincerely,

Mareile Wolff

Special Issue Editor

**Response to Reviewer #1**

Thank you Dr. Bardsley for reviewing the manuscript and offering some insightful and helpful comments for improvements. We have addressed each below and will incorporate these changes.

Comments on:

*Measuring precipitation with a geolysimeter*

By C.D. Smith et al.

General:
The paper is a useful contribution to the literature, in this case with the novelty aspect of using the area-integrating capabilities of geolysimeters as a snow measure to offset in part the high spatial variability of snow amount.

My comments are all of a minor nature related to the rainfall aspect, but might be taken into consideration in a revision.

Page 2 line 25

*However, the precipitation comparisons done so far have been more qualitative than quantitative due to the spatial separation of the geolysimeter and the measuring precipitation gauges.*

As noted in the independent comment, for the Bardsley-Campbell geolysimeter there was never at any time a spatial separation of the geolysimeter and the rain gauge. Is it really the case that in all other cited geolysimeter studies there was spatial separation between rain gauge and geolysimeter?

We thank the reviewer for providing an example of a quantitative intercomparison. Much of the literature cited as examples of intercomparisons have been largely qualitative which result from a spatial separation between the measurements. We have now used the Bardsley and Campbell (2007) example in New Zealand as an exception. The statement on page 3 of the manuscript has been updated to read: "Both Bardsley and Campbell (2007) and Barr et al. (2000) report a close correspondence between co-located geolysimeter and precipitation gauge measurements but do not include detailed quantitative analysis of this correspondence. Previous intercomparisons also do not include a discussion on measurement of snowfall. "

Top of page 7.

Were the 30 minute well water levels recordings of the well water levels every 30 minutes, or the average of higher time resolution monitoring from the previous 30 minutes?

The water level was sampled at the beginning of every 30 minute period and were not an average of high frequency measurements made over the previous 30 minute period, as a temperature measurement would be made. The text in the Methods section was updated to read "The raw 30 min deep well observations, sampled at the beginning of each 30 minute period, require an adjustment for the effects…" to clarify.

Fig. 2

The slope change in water level following the rain event in Fig. 2a is certainly a good argument for site discharge loss being the cause of the evident slight under-estimation of the rain event. However, is this possibly a time resolution effect? That is, the pore water pressure increase from surface loading is instantaneous for practical purposes but there will be some finite time (a few minutes?) before the site rainwater starts to depart as discharge after a sudden event. What was the duration of the rainfall event in Fig. 2a? If it was less than 30 minutes and water levels were recorded every 30 minutes, is it possible for the maximum rise of water level to have been missed due to the relatively coarse sampling interval?

This is a very good point and we will make a note of this in the Discussion as a source of error. We could also suggest higher frequency measurements in future intercomparisons to try to quantify this error. Unfortunately, these measurements were not available here.

The text added on page 11 is as follows: "Another consideration that has a potential impact on the timing and magnitude of the precipitation geolysimeter estimates as shown in Fig. 2b is the temporal resolution of the geolysimeter observations. Because the response of the geolysimeter to rainfall loading is nearly instantaneous, the minimum or the peak water level in the well may have been missed by the water level readings that were taken once every 30 minutes thus resulting in an under estimate of precipitation. This effect would only be significant if water losses from the geolysimeter response area by runoff or evapotranspiration were significant during the 30 minutes before or after the beginning or end of the precipitation event. Considering the low relief of the study area, runoff is slow (cf Fig 2a) and the error due to the sampling interval is likely to be much smaller than 1 mm."

 Some comment might also be added as to the likely effect (or not) of 1 km spatial variation of rainfall, given that precipitation is a point measure and the geolysimeter is a spatial average. It is a pity that at least one more precipitation gauge was not in operation at the site, but presumably other measurements of closely spaced gauges in similar environments might be mentioned in this respect.

This was a point raised by another reviewer and is addressed in more detail in our comments posted there.

Some comment should be made about the cause of the declining trend in water level prior to the rain event – evaporation and / or net groundwater export from the site?

Certainly. Although it is difficult to ascertain if the decline in water level is a result of evaporation or groundwater loss from the response area, this decline largely occurs between noon and 10pm local time so we can assume that most is evaporation. The following sentence on page 8 in the Results section was modified to read: "Evapotranspiration was likely very small since relative humidity during the night and

following the precipitation event was 100 %. The decline in water level in the well from 18:00 UTC (12:00 local) to 4:00 UTC (22:00 local) prior to the event is likely indicative of evapotranspiration. "

The zero point of the rainfall plot should be set to correspond with the start of the rainfall event

This is a good suggestion and we have updated Fig 2 accordingly.

The rainfall representation in both plots of Fig 2 is confusing and should be converted to cumulative rainfall (no negative slopes).

The raw cumulative gauge data may at times show negative slope, as shown in Fig 2. However, the smoothed and filtered cumulative gauge record does not have negative slopes, because, as pointed out by the reviewer, cumulative rainfall can only increase, by definition. Fig 2 has been updated accordingly. To clarify these plots further, we have also added the filtered and accumulated well data. Now the graphs show both the raw well data (which illustrates the evaporation) but also the filtered and accumulated data which is used for intercomparison with the filtered and accumulated precipitation data.

**Response to Reviewer #2**

I would like to thank the anonymous reviewer for taking the time to thoroughly review this manuscript and offer suggestions for revisions. The comments are appreciated and will serve to improve the manuscript.
This paper could make a useful contribution by quantifying the relationships between a precipitation gauge and a geolysimeter. The authors have a done a good job of identifying many of the hydrological processes which can account for some of the differences between the sets of measurements, particularly those of the snowfalls.

Unfortunately, the authors have not adequately accounted for the difference between the areas of the rain gauge and the geolysimeter. The areal reduction factor, which quantifies the reduction of rainfall extremes over a region, compared to a point, is well known in hydrology. ARF values have been derived for many regions and are a standard part of engineering hydrology. Because the area of the geolysimeter is so large (almost 5 $km^2$) it approaches the sizes of the regions referenced in some published areal-reduction factor curves.

More theoretical analyses (De Michele et al, 2001, among others) also demonstrate that the reduction factor is related to the size of an event, which is also shown by the plot of the geolysimeter and gauged rainfalls in Figure 3. However, the reduction factor also depends on the length of the event, while the authors have combined events of varying lengths. It would be possible to compare areal reduction factors for intensities, durations and frequencies derived from the data with published values.

At the very least, the effect of the area of the geolysimeter on the difference between its rainfall estimates and those of the gauge needs to be addressed.

De Michele, Carlo, Nathabandu T. Kottegoda, and Renzo Rosso. "The derivation of areal reduction factor of storm rainfall from its scaling properties." Water Resources Research 37, no. 12 (2001): 3247-3252

The reviewer brings up a good point that we had originally only considered lightly. Quantifying event lengths is somewhat of an arbitrary methodology with a very wide variety of thresholds used in the literature, so we would rather avoid analysis based on quantified event length for this manuscript but would reconsider further analysis in the future. In this analysis, events have been identified that have a clear beginning and clear end to precipitation, (as shown in the example in Figure 2) and if you define the event length to start at the first trace of precipitation and end immediately after the last trace of precipitation with no breaks larger than a few hours in between, then our lengths vary from 7 to 74 hours, averaging about 24 hours. Extrapolating from De Michele et al. (2001), the approximate ARF for the average event length is about 95%. Applying this adjustment, the bias between the geolysimeter and

the gauge is reduced to -1.5% from -7.0% and the rmse is reduced to 1.6 mm from 2.3. The slope of the regression line also becomes closer to 1 (increasing from 0.90 to 0.95). Therefore, precipitation scaling could certainly explain much of the negative bias in the geolysimeter, combined with evapotranspiration, runoff, and potential timing of geolysimeter peaks relative to the timing of the measurement. However, we feel that areal reduction factors for this field site are neither well known nor understood. This is complicated by the fact that the geolysimeter is at the centre of the response area and the sensitivity to load changes per unit area falls off with distance, making us less confident in using any ARF.

This is addressed in the Discussion section on page 11.

General The writing needs revision. The language is excessively colloquial and the terminology is frequently sloppy. A few examples are shown below

Besides the corrections noted below, we will do another thorough proofread and correct language issues.

Page 1, Line 15 "Correlations varied from 0.99 for rainfall to 0.94 for snowfall." I believe that you are referring to the correlation coefficients of the linear regressions ($r^2$) rather than values of correlations between the data sets.

Corrected

P 3, L 3 "wider area" Area is not the same thing as width! This sloppy usage is repeated throughout the document."(hectares vs $m^2$)" The exact areas of the gauge orifice and of the geolysimeter and their ratio should be given. This sentence grossly understates the ratio, i.e. the ratioof 1 hectare to 1 m2is 10,000:1. According to the manufacturer's website, the gauge orifice area is 200 $cm^2$, i.e. 0.02 $m^2$. If the radius of the geolysimeter measurement
area is 1.25 km (as stated), then the ratio is more than 245 million to 1!

The intent of this analysis was not to do an extensive scaling experiment. We don't want to debate the spatial representativeness of the 200 $cm^2$ gauge orifice. The point of this sentence was simply to state that there could be advantages to measuring precipitation at the scale of hectares rather than square metres (or smaller). The reference to "$m^2$" has been removed to avoid the implication that this is a scaling experiment. The language has been corrected in the rest of the manuscript.

P 4, L 7 "This stress transmission" The previous sentence refers to the load (i.e. a force) and the pore-water pressure, not to a stress. Please make this clearer.
L 27 "at 13U 417810E, 5863437N." Why not specify the location by its longitude and latitude? They are global values, rather than being specific to a region, and are more easily understood.

Corrected

P 7, L 2 "and earth tides"
Earth tides were not mentioned previously, when discussing the adjustment of the raw data, but probably should have been.

This was included in the context of the methodology. A mention of the earth tide effect is now included in the Groundwater theory section (Page 5, lines 6-10), including a reference.

P 8, L 13 "Evapotranspiration was likely minimal since relative humidity during the night
..."

Does plant transpiration of water ever occur at night? The word "minimal" is being used in a colloquial sense. It would be better to say "very small".

Plant transpiration can occur at night and so can evaporation. We qualitatively state that these are likely minimal because humidity is high and therefore vapour pressure deficit is likely small, but we don't quantify the evapotranspiration so this is speculative. We will change the wording as suggested.

L 21 "significant" This word should not be used in a scientific paper, unless you are giving the level of significance.

Agreed

P 9, L 1 "Summary statistics ..." How were these computed? What program did you use?

MATLAB

L 5 "RMSE varies ..." The abbreviation should be defined. Also, since the gauge data are also believed to be in error, what you are actually computing is the root mean squared deviation (RMSD) between the two datasets.

OK, changed RMSE to RMSD and defined the abbreviation.

P 10, L 29 What is an "adequate snow catchment"?

Poor choice of words. The sentence reads correctly when the word "adequate" is removed.

Figures P 18 Figure 1 caption "response area of ~1.25 km" Area is not measured in km. This would appear to be the radius of the geolysimeter response area, correct?

This has been corrected to read "...with a radius of ~1.25 km".

P 24 Figure 7 It appears that a point is missing from the plot. There is a point plotted for largest gauge unadjusted precipitation, and for the sigmoidal adjusted value, but there does not appear to be a corresponding point for the exponential arctan adjusted value.

Those plot axes were scaled incorrectly. This was corrected.